# Personal Cell Phones among Children: Parental Perception of Content-Related Threats and Attempts to Control Them in a Lithuanian Sample

**DOI:** 10.3390/bs12060185

**Published:** 2022-06-09

**Authors:** Donatas Austys, Ausma Sprudzanaitė, Rimantas Stukas

**Affiliations:** Department of Public Health, Institute of Health Sciences, Faculty of Medicine, Vilnius University, M. K. Čiurlionio 21/27, LT-03101 Vilnius, Lithuania; ausma.sprudzanaite@santa.lt (A.S.); rimantas.stukas@mf.vu.lt (R.S.)

**Keywords:** cell phones, children, parents, attitudes, risk factors

## Abstract

*Background and Objectives*: Various content-related threats such as provocative content, disinformation, cyberbullying, or sexual and commercial messages might reach children by phone and have a negative effect on their health. Amongst parents who are able to control it, we aimed to assess parental attitudes towards the use of mobile phones among children and control measures taken. *Materials and Methods*: A total number of 619 parents of primary school children from a middle-sized town in Lithuania participated in this study. Parents anonymously filled out our original questionnaire. Distribution of the respondents was assessed according to control measures taken, threat awareness, and sociodemographic factors. *Results*: Most of the respondents (79.8%) thought that personal mobile phones might be harmful to children’s health, 99.5% of the parents used at least one control measure. Further, 91.9% of the respondents did not think that children might receive messages from strangers. Respectively, 85.5% and 95.2% of the parents thought that children do not receive offensive or sexual content messages. Many parents (25.5%) helped their children register to social networks. Parents with lower education and parents of younger children had lower awareness of threats (*p* < 0.05). Fathers, higher educated, single, and unemployed parents indicated application of fewer control measures (*p* < 0.05). Other sociodemographic factors were not related with threat awareness or control measures taken (*p* > 0.05). *Conclusions*: Nearly all parents of primary school children take measures in order to control their children’s usage of mobile phones but most of them underestimate content-related threats brought by mobile phones.

## 1. Introduction

Results of this study were presented at the 16th World Congress on Public Health [1].

New information and communication technologies are becoming increasingly more prevalent among children every year [2,3]. Studies show that children become users of personal mobile phones approximately when they turn 9 years old, and every second child aged from 9 to 16 years old uses a mobile phone to contact parents daily [4]. Mobile phones, especially smartphones, bring opportunities for parental control [5], crisis management, including helplines for suicide prevention [6], mobile health, and real-time surveillance applications [7]. Despite many possible advantages, mobile phones bring threats as well. Studies show that every second child in the age group from 9 to 16 years uses smartphone to access the internet every day [4]. This brings various threats including content-related (such as provocative content or wrong information), contact-related (such as cyber bullying or sexting), and commercial-related (such as commercial exploitation or collection of personal data without informed consent) [8].

Studies show that adolescents tend to perceive the vast majority of risky behavior using modern information and communication technologies less frequently than adults [9]. Among teenagers, sexting is prevalent with intentions of flirting, sexual experimentation, or expression of a desire to start sexual relationships [10]. According to studies, a portion of primary and secondary school pupils (from 11 to 17 years old) send and share sexual content. This might lead to cyberbullying, which is common among children. It was found that approximately 33% of primary and secondary school pupils suffer from verbal attacks, 17% from threats and intimidation, 10% from identity theft, 7% from blackmail, and up to 11% are embarrassed by spreading visual content [11]. Studies show that cyberbullying is associated with post-traumatic stress disorder (PSTD), depression, and suicidal thoughts [12,13]. In order to protect children, parents use restrictions related with time, device, content, location, and purchases. Additionally, it has been shown that some parents try to supervise children without strict restrictions and let them use mobile phones independently or decide not to intervene and give autonomy to their children [14]. However, there is still a lack of research about parental attitudes towards personal mobile phones usage among children. While parents are able to control many aspects of children’s lives, we aimed to disclose parental attitudes towards the use of mobile phones among children.

## 2. Materials and Methods

### 2.1. Characterization of the Data Collection Procedure

The study was conducted in the three largest schools of a medium-sized town of Lithuania (from 10,001 to 100,000 residents) in March 2019. The participants of this study were parents of children attending primary classes in those schools. Every parent associated with those schools was invited to participate in our study and to fill out our original questionnaire about parental attitudes towards personal cell phone usage among children. Every respondent participated in this study anonymously. Before the survey, participants were informed that participation in this study is voluntary and that no personal data which could be used for direct identification is required. In order to ensure anonymity and confidentiality, filled out questionnaires were collected blindly, and the sample size was big and results were presented in an aggregated way. In total, 643 parents agreed to participate in our study and completed the questionnaire. After the exclusion of 24 questionnaires with half or more of the questions left unanswered, a total of 619 parents’ responses were used for statistical analysis.

### 2.2. Description of the Questionnaire

The anonymous original questionnaire consisted of 30 close-ended questions and 3 open-ended questions. In this paper, we used part of the questionnaire. All questions used for this paper except the social and demographic characteristics of the respondents and their children are described in Table 1. The questions about the social and demographic characteristics of the participants included gender, age, education, type of place of residence, marital status, employment status, and income. The questions about children of the respondents included children’s gender and age.

### 2.3. Statistical Analysis

We used questions about threats associated with personal mobile phone usage among children to make a categorical variable in order to categorize parents into two groups: those who were aware about the risks to children’s health arising from the usage of personal mobile phones and those who were not aware of such risks. Firstly, we calculated the number of marked answer options indicating the awareness of threats to children’s health (Table 1). Then, we used this number to make a binary variable: respondents who marked from 0 to 4 threats were assigned to the low awareness group and respondents who marked from 5 to 8 threats were assigned to the high awareness group. Additionally, we used questions about measures (time control, restriction of internet access, or strict rules) applied by parents to make a binary variable (Table 1). We categorized parents into two groups: those who apply all 4 control measures mentioned in our questionnaire were assigned to a group of parents who apply more control measures, others we assigned to a group where parents apply fewer control measures.

In order to simplify the interpretation of the analysis, all social and demographic characteristics of the respondents were used as binary variables. The respondents who graduated from primary, lower secondary, secondary or vocational school were assigned to the lower education category. Respondents who graduated from technical school, studied and/or finished studies in a college or university were assigned to the higher education category. The respondents from small towns (from 3001–10,000 residents), villages with 501–3000 residents, and villages up to 500 residents were assigned to one group and those living in a city with more than 10,001 residents were assigned to another group. The group of single respondents included those who indicated that they are single, as well as widows and widowers and divorced respondents. The respondents who indicated that they are CEOs or owners of a company, civil servants, employees in a service sector/salesmen, office workers, specialists, or farmers were assigned to the employed respondents’ group, those who indicated that that they are retirees, on parental leave, or unemployed were assigned to the unemployed respondents group.

Normality of the variables’ distribution was tested using the Shapiro–Wilk test. With respect to the results of this test, medians with interquartile range (Q1–Q3) were presented for variables with non-normal distribution, and averages with standard distributions were presented for variables with normal distribution. Pearson’s chi-squared test (χ2) was used to determine whether there was a statistically significant difference between the expected frequencies and the observed frequencies in one or more categories. Differences were considered statistically significant when *p* < 0.05.

## 3. Results

The response rate to our survey was high and accounted for 96.3%; only a small part (3.7%) of the data was missing. Relative frequencies further in this paper are presented excluding responses with missing answers.

The majority of the participants in this study (86.4%) were females. The median age of the respondents was 37 (range 24–59) years. Residents from urban and rural areas, as well as residents with higher and lower education, took up similar portions of the sample. Most of the participants were married (69.1%), employed (73.2%), and with a monthly average income (net) per family member 301 euros or more (62.0%). The median age of the children was 9 (range 8–10) years.

The majority of the respondents thought that children do not receive messages from strangers and do not communicate with such persons. Additionally, most of the participants of this study thought that children do not receive offensive or sexual content messages. Only one-fifth of the parents (20.2%) thought that mobile phones are not harmful to children’s health; one-eighth of the parents (11.8%) indicated that their children were under 6 years old when they started to use personal mobile phones. About a quarter of the respondents (25.5%) helped their children to create accounts on social networks despite age restrictions; less than a half of the parents thought that their children were active users on social networks (Table 2).

According to the distribution of the respondents by the number of indicated threats to children caused by the usage of their personal mobile phones, approximately half of the respondents (54.9%) indicated 5 or more possible threats and were assigned to the group of the respondents with high awareness of threats. A similar proportion (45.1%) of the parents indicated 4 or fewer threats and were assigned to the group of the respondents with low awareness of threats. Only three respondents (<1.0%) indicated no possible threats, and only one person indicated all 8 out of 8 possible threats. One, two, three, four, five, six, and seven threats were indicated by 0.8%, 3.2%, 7.3%, 33.3%, 28.4%, 19.9%, and 6.5%, respectively.

Although most of the parents indicated that they try to control their children’s usage of mobile phones, a large part (30.0%) of the respondents indicated permission to use mobile phones for an average of three hours or more per day, 9.4% of the sample indicated no conversations with children about the functions of their mobile phones they use, 7.8% reported no control of their children’s usage of personal mobile phones, and 19.2% claimed no control of their children communication contacts. According to the distribution of the respondents by the number of control measures taken, most of the parents indicated 3 or 4 control measures (respectively, 39.7% and 48.9% of the sample). A small part of the respondents indicated only 1 or 2 control measures (respectively, 3.1% and 7.8% of the sample). Very few (0.5%) respondents indicated using no control measures.

According to the distribution of the respondents by the awareness of threats and control measures taken, more of the parents with a high awareness of threats indicated no control of children’s usage of personal mobile phones (*p* = 0.009) and permission for children to use mobile phones for an average of three hours or more per day (*p* = 0.005). On the other hand, more of the parents with a high awareness of threats indicated control of children’s communication contacts (*p* < 0.001) (Table 3).

According to the distribution of the respondents by the awareness of possible threats and social and demographic factors, a larger part of the parents with higher education had a high awareness in comparison with the group of parents with lower education (*p* = 0.021). Additionally, parents of younger children more frequently had a low awareness than parents of older children (*p* = 0.003). Gender, age of the respondent, place of residence, marital status, employment status, income, and child’s gender were not related to the awareness of the possible threats (*p* > 0.05). According to the distribution of the respondents by the absence of the control measures taken and social and demographic factors, fathers less frequently indicated application of the control measures than mothers (*p* = 0.032). Additionally, higher educated, single, and unemployed parents indicated application of fewer control measures than lower educated, married, and employed parents (*p* values were, respectively, 0.013, 0.019, and 0.041) (Table 4).

## 4. Discussion

Results of this study show that most of the parents of primary school pupils think that personal mobile phones might be harmful to children’s health. Additionally, the results show that nearly all parents use various control measures (such as control of mobile phone functions availability, communication contacts, and time) to prevent the possible harm to children’s health. Despite this, the vast majority of parents do not pay attention to threats brought by communication and social networking by mobile phones. Although most of the respondents of our study underestimated such threats as communication with strangers and receiving offensive or sexual content messages, other studies show that children actually face such risks and sometimes suffer from them [15,16,17,18]. Furthermore, many parents help their children to register to social networks, ignoring the age restrictions and possible threats brought by social networking. Studies performed during the COVID-19 pandemic showed that children’s and youths lifestyle changed and screen time increased [19,20]. This may lead to an increase in the prevalence of risky behavior online. These results show the lack of parental knowledge about such threats and might be related with an emphasis on other possible threats of mobile phones that were not included in our questionnaire. According to the World Health Organization (WHO), research does not suggest any consistent evidence of adverse health effects from exposure to radiofrequency fields of mobile phones [21].

Eighty percent of the respondents of our study did not think that children may experience cyberbullying. This could mean that most of the parents do not understand the potential extent and risk to children’s health of cyberbullying. Bottino et al.’s systematic review showed that the prevalence of cyberbullying among youth aged from 10 to 17 years old ranged from 6.5% to 35.4% [18]. John et al.’s systematic review revealed that children and young people who have experienced cyberbullying are 2.35 times more likely to self-harm, 2.57 times more likely to attempt suicide, and 2.15 times more likely to have suicidal thoughts than individuals who have not experienced cyberbullying [22]. Additionally, more than ninety percent of the parents who participated in our study did not think that children may receive sexual content messages. Barrense-Dias et al.’s review revealed that prevalence rates of sexting among children ranged from 0.9% to 60% [17]. Kosenko et al.’s meta-analysis showed weak associations between engaging in sexting and risky sexual practices. Multiple studies showed that sexting might be associated with problematic behaviors such as intimate partner violence [23,24,25]. Additionally, studies show that social networking might be associated with body image concerns, eating disorders, and negative mood after browsing Facebook [26,27]. About a quarter of the respondents of our study helped their children to create accounts on social networks despite age restrictions and these threats.

In most cases, parents play the main role in children’s social life. Studies show that parental mediation influences children’s use of the internet and social networks by inducing appropriate online behaviors and preventing cyberbullying [28], contact with strangers [29], and online harassment [30]. Participants of our study indicated the use of at least a few of control measures. Only a few parents indicated taking no control measures. Bybee et al. presented a three-dimensional model of mediation. It included restrictive mediation, active mediation, and co-use [31]. Nikken et al. proposed to extend the three-dimensional model to five dimensions by dividing the restrictive mediation dimension into time restriction and special content restriction, adding supervision as a new dimension [32]. However, it was also noted that parental mediation can cause the “Pandora effect” which emphasizes that an increase in parental monitoring may lead to an increase in the problematic use of mobile phones by teenagers [33]. Surprisingly, our study showed that more of the parents who had higher awareness of threats indicated no control of children’s usage of personal mobile phones and permission to use mobile phones for an average of three hours or more per day. Further studies are needed in order to investigate whether fewer control measures might reduce the incidence of problematic use of mobile phones by children.

According to the results of our study, parents might not be aware about many content-related threats brought to children by their usage of mobile phones. Therefore, it would be beneficial to inform parents about risks to children’s health brought by their personal mobile phones. Schools provide an advantageous environment to perform this. As stated by Baraldsnes D, in order to prevent cyberbullying, pupils, parents, and all school personnel should be taken into account. According to her, teachers need to learn how to identify and handle cyberbullying incidents. The need of supervision when technologies are used was emphasized [34]. Additionally, higher levels of family and teacher social support [35], talking about risks that arise due to usage of mobile phones, also talking about healthy relationships are shown to be beneficial [36]. In addition, the importance of training for teachers is emphasized. Inclusion of training in cyberbullying to the pedagogical study programs is suggested [37].

This study had several limitations. First of all, this study should be deemed exploratory and needs to be replicated in the future. Due to a large number of statistical tests, control for alpha inflation should be taken into account. Secondly, children being active users in social networks was considered a threat because it may show an increase in time spent using mobile phones and especially an increase in time children might be exposed to the threats of social networks. However, this answer might also show a lower risk to children due to possibly better parental understanding of children’s activities using mobile phones. Additionally, helping children to create accounts in social networks despite age restrictions was considered a threat in this study, however, it might be considered oppositely because, in that case, details of children’s social network accounts are known to parents, and this might provide better parental control. In addition, although this study was conducted in a medium-sized town in Lithuania and included a sufficient sample of parents of primary-school-aged pupils, the results of this study may not reflect the actual level of threat awareness and control measures of the whole Lithuanian population of parents. Additionally, despite the fact that, in order to clearly present the results of our study, we made categorical variables for threat awareness and control measures applied, cut-off points used to make categories from continuous variables might not be optimal. Therefore, this study should be replicated allowing these variables to be retained as continuous variables in future analyses, also addressing other mentioned limitations.

## 5. Conclusions

Nearly all parents of primary school children take measures in order to control their children’s usage of mobile phones, but most of them underestimate content-related threats brought by mobile phones.

## Figures and Tables

**Table 1 behavsci-12-00185-t001:** Questions of the questionnaire about threats associated with personal mobile phones usage among children and parental control measures taken in order to control children’s usage of mobile phone.

**1. Questions about threats associated with personal mobile phone usage among children**	
Question	Answer options
Who, in your opinion, do children (not necessarily yours) get messages from?	From friends/From classmates/From teachers/From family members/From strangers *
If you think that children (not necessarily yours) get messages from strangers, do you think that children communicate with them?	Yes */No
In your opinion, what content of messages do children (not necessarily yours) receive by phone?	Informational content/Friendly content/Offensive content */Promotional content/Sexual content *
In your opinion, could mobile phones be dangerous for children’s (not necessarily yours) health?	Yes/No *
When did your child start to use his/her personal mobile phone?	An open-ended question was categorized into the range under 6 years old * and 6 years old or older
If your child uses social networks, how did he/she start to use them?	We helped to register*/Other family members helped to register/Friends helped to register/Registered by himself (herself)
In your opinion, does your child actively use functions of social networks (e.g., sharing photos, videos, sends textual messages)?	Yes/No *
**2. Measures applied by parents in order to control mobile phone usage among their children**	
Questions	Answer options
Does your child talk to you about the mobile phone functions he/she uses?	Yes, we frequently talk about it/No, we do not talk about it**/I am not interested**
Do you control your child’s usage of his/her personal mobile phone?	Yes, we control the time our child spends using the phone/Yes, we restrict an internet access/Yes, we have strict rules/No, we do not interfere **
What is the average time your child uses his/her personal mobile phone per day?	Less than 1 h/1–2 h/3–4 h **/5–6 h **/7 h and more **
Do you control who your child communicates with?	Yes/No **

* Answer options indicating threats to children’s health; ** answer options indicating the absence of measures applied to control risks to children health arising from the usage of personal mobile phones.

**Table 2 behavsci-12-00185-t002:** Distribution of the respondents by possible threats to children caused by the usage of their personal mobile phones (*n* = 619).

Threats	Cases	Relative Frequency
Thinking that children do not receive messages from strangers	569	91.9%
Thinking that children do not communicate with strangers	606	97.9%
Thinking that children do not receive offensive content messages	526	85.0%
Thinking that children do not receive sexual content messages	589	95.2%
Thinking that mobile phones are not harmful to children’s health	125	20.2%
Using a personal mobile phone under 6 years old	73	11.8%
Helping children to register to social networks	158	25.5%
Thinking that children are active users in social networks	264	42.6%

**Table 3 behavsci-12-00185-t003:** Distribution of the respondents by control measures taken and awareness of threats.

Variable	Low Awareness of Threats	High Awareness of Threats	*p*-Value
**Does your child talk to you about the mobile phone functions he/she uses?**			
Yes we frequently talk about it	45.5%	54.5%	0.553
No, we do not talk about it/I am not interested	41.4%	58.6%	
**Do you control your child’s usage of his/her personal mobile phone?**			
Yes (time control, restriction of internet access, or strict rules)	46.6%	53.4%	0.009 *
No, we do not interfere	27.1%	72.9%	
**What is the average time your child uses his/her personal mobile phone per day?**			
2 h or less	48.7%	51.3%	0.005 *
3 h or more	36.3%	63.4%	
**Do you control who your child communicates with?**			
Yes	40.2%	59.8%	<0.001 *
No	65.5%	34.5%	

* Statistically significant difference.

**Table 4 behavsci-12-00185-t004:** Distribution of respondents by control measures taken, threat awareness, and sociodemographic factors (*n* = 619).

Variable	Low Awareness of Threats	High Awareness of Threats	*p*-Value	Fewer Control Measures	More Control Measures	*p*-Value
**Gender**						
Female	44.3%	55.7%	0.329	49.3%	50.7%	0.032 *
Male	50.0%	50.0%		61.9%	38.1%	
**Age**						
37 years old or younger	45.0%	55.0%	0.948	49.6%	50.4%	0.405
38 years old or older	45.2%	54.8%		52.9%	47.1%	
**Education**						
Lower education	49.0%	51.0%	0.021 *	46.8%	53.2%	0.013 *
Higher education	39.7%	60.3%		56.9%	43.1%	
**Place of residence**						
Medium-sized city	47.7%	52.3%	0.208	49.0%	51.0%	0.322
Small town, village	42.6%	57.4%		53.0%	47.0%	
**Marital status**						
Married	45.1%	54.9%	0.988	47.9%	52.1%	0.019 *
Single	45.0%	55.0%		58.1%	41.9%	
**Employment**						
Employed	46.6%	53.4%	0.214	48.6%	51.4%	0.041 *
Unemployed	41.0%	59.0%		57.8%	42.2%	
**Income**						
300 EUR or less	44.1%	55.9%	0.658	55.9%	44.1%	0.087
301 EUR or more	46.0%	54.0%		48.4%	51.6%	
**Child’s gender**						
Girl	43.2%	56.8%	0.342	51.7%	48.3%	0.769
Boy	47.0%	53.0%		50.5%	49.5%	
**Child’s age**						
From 6 till 9 years old	49.4%	50.6%	0.003*	51.6%	48.4%	0.708
From 10 till 12 years old	36.7%	63.3%		50.0%	50.0%	

* Statistically significant difference.

## Data Availability

Not applicable.

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
