# Peer review of "Personal Cell Phones among Children: Parental Perception of Content-Related Threats and Attempts to Control Them in a Lithuanian Sample"

_behavsci, 2022, doi:10.3390/bs12060185_

Round 1

Reviewer 1 Report

Dear Authors,  Please find below my comments:

The aim of the study is to assess parental attitudes towards use of mobile phones among children and control measures taken

Research: the research  was correctly designed

Methods: the methods are well presented and described

Results and Conclusions:

 The results are well presented 

Conclusions (es below)  is not clear, and some corrections of description are needed , its like to say: Majority of the parents  are aware of the threats , but most of the parents are not aware of the threats…what kind of conclusion it is? What does mean Majority of parents? Most of the parent? It means how many? In  my opinion the conclusions should be reformulated inluded abstract. In the conlusions is also a typo in  word : "but", in the text and in abstract is “bus”

„Majority of parents of primary school children think that mobile phones might be  harmful to children’s health and take measures in order to control it bus most of parents  to do not pay attention to content-related threats brought by phones.”

Other: The topic is very important for pedagogical the topic is important for the pedagogical community, and in my opinion the obtained results of the research also require the authors to indicate pedagogical recommendations (implications for educational practice), it is about the necessity of teachers' actions to raise the awareness of parents on this subject 

Author Response

Point 1: Conclusions (es below)  is not clear, and some corrections of description are needed , its like to say: Majority of the parents  are aware of the threats , but most of the parents are not aware of the threats…what kind of conclusion it is? What does mean Majority of parents? Most of the parent? It means how many? In  my opinion the conclusions should be reformulated inluded abstract. In the conlusions is also a typo in  word : "but", in the text and in abstract is “bus”

„Majority of parents of primary school children think that mobile phones might be  harmful to children’s health and take measures in order to control it bus most of parents  to do not pay attention to content-related threats brought by phones.”

Response 1: Thank you for your review. We reformulated the conclusions.

Point 2: Other: The topic is very important for the pedagogical community, and in my opinion the obtained results of the research also require the authors to indicate pedagogical recommendations (implications for educational practice), it is about the necessity of teachers' actions to raise the awareness of parents on this subject

Response 2: We did not assess the efectiveness of interventions in this study. Therefore, we could not provide exact recommendations about effective measures that could be taken by teachers. On the other hand, we added a paragraph about possible implications for educational practice to the discussion section emphasizing the importance to inform parents about threats brought to children by their usage of personal mobile phones.

Reviewer 2 Report

Dear authors. I found your manuscript interesting, touching important issue and well structured. Still I would suggest to improve some parts.

  1. Information about ethical issues is missing. What was the procedure of the informed consent? Was your study reviewed by local Ethical board? What confidentiality measures were taken?
  2. I would suggest to use the same terms through the text. E.g. use "low awareness vs high awareness group" instead of changing into "perceive threats well vs worse".
  3. I would suggest to reduce the number of the tables, leaving some information in the text, or avoiding excessive information, e.g. tables 3,5,7. 

Best wishes.

Author Response

Point 1: Information about ethical issues is missing. What was the procedure of the informed consent? Was your study reviewed by local Ethical board? What confidentiality measures were taken?

Response 1: Thank you for your review. We added the missing information to the manuscript.

Point 2: I would suggest to use the same terms through the text. E.g. use "low awareness vs high awareness group" instead of changing into "perceive threats well vs worse".

Response 2: We replaced “perceive threats well/worse” with “high/low awareness of threats”.

Point 3: I would suggest to reduce the number of the tables, leaving some information in the text, or avoiding excessive information, e.g. tables 3,5,7.

Response 3: We reduced the number of tables by deleting table 3, table 4 and table 5 and adding some additional explanatory text in order not to lose information which was provided in those tables. We did not remove the table 7 as it provides the generalised results of this study and removal of it, in our opinion, would not be beneficial.

Round 2

Reviewer 1 Report

Dear Authors,  Now the paper after all changes has much better quality.  I accept you proof and answers for my notes.Thank you from cooperation

Author Response

Point 1: Dear Authors,  Now the paper after all changes has much better quality.  I accept you proof and answers for my notes.Thank you from cooperation

Response 1: Thank you for your review and valuable suggestions in order to increase the quality of our paper.